# Atomistic characterization of the active-site solvation dynamics of a model photocatalyst

Tim B. van Driel[1,2,*], Kasper S. Kjær[1,3,4,*], Robert W. Hartsock[3], Asmus O. Dohn[5,†], Tobias Harlang[1,4], Matthieu Chollet[2], Morten Christensen[1], Wojciech Gawelda[6,7], Niels E. Henriksen[5], Jong Goo Kim[8,9], Kristoffer Haldrup[1], Kyung Hwan Kim[8,9,†], Hyotcherl Ihee[8,9], Jeongho Kim[10], Henrik Lemke[2,†], Zheng Sun[3], Villy Sundström[4], Wenkai Zhang[3,†], Diling Zhu[2], Klaus B. Møller[5], Martin M. Nielsen[1] & Kelly J. Gaffney[3]

The interactions between the reactive excited state of molecular photocatalysts and surrounding solvent dictate reaction mechanisms and pathways, but are not readily accessible to conventional optical spectroscopic techniques. Here we report an investigation of the structural and solvation dynamics following excitation of a model photocatalytic molecular system $[Ir_2(dimen)_4]^{2+}$, where dimen is para-diisocyanomenthane. The time-dependent structural changes in this model photocatalyst, as well as the changes in the solvation shell structure, have been measured with ultrafast diffuse X-ray scattering and simulated with Born-Oppenheimer Molecular Dynamics. Both methods provide direct access to the solute–solvent pair distribution function, enabling the solvation dynamics around the catalytically active iridium sites to be robustly characterized. Our results provide evidence for the coordination of the iridium atoms by the acetonitrile solvent and demonstrate the viability of using diffuse X-ray scattering at free-electron laser sources for studying the dynamics of photocatalysis.

[1] Molecular Movies, Department of Physics, Technical University of Denmark, DK-2800 Kongens Lyngby, Denmark. [2] LCLS, SLAC National Accelerator Laboratory, Menlo Park, California 94025, USA. [3] PULSE Institute, SLAC National Accelerator Laboratory, Stanford University, Stanford, California 94305, USA. [4] Chemical Physics Department, PO Box 124, Lund University, S-22100 Lund, Sweden. [5] Department of Chemistry, Technical University of Denmark, DK-2800 Kongens Lyngby, Denmark. [6] European XFEL GmbH, Holzkoppel 4, D-22869 Schenefeld, Germany. [7] Institute of Physics, Jan Kochanowski University, 25-406 Kielce, Poland. [8] Department of Chemistry, KAIST, Daejeon 305-701, South Korea. [9] Center for Nanomaterials and Chemical Reactions, Institute for Basic Science, Daejeon 305-701, South Korea. [10] Department of Chemistry, Inha University, Incheon 402-751, South Korea. * These authors contributed equally to the work. † Present addresses: Science Institute of the University of Iceland, VR-III, 107 Reykjavik, Iceland (A.O.D.); Department of Physics, AlbaNova University Center, Stockholm University, SE-10691 Stockholm, Sweden (K.H.K.); Paul Scherrer Institut, WSLA/207, 5232 Villigen PSI, Switzerland (H.L.); Center for Advanced Quantum Studies, Department of Physics, Beijing Normal University, Beijing 100875, China (W.Z.). Correspondence and requests for materials should be addressed to T.B.v.D. (email: timbvd@slac.stanford.edu) or to K.S.K. (email: kaspersk@gmail.com) or to K.B.M. (email: kbmo@kemi.dtu.dk) or to M.M.N. (email: mmee@fysik.dtu.dk) or to K.J.G. (email: kgaffney@slac.stanford.edu).

The efficiency, selectivity and rate of chemical reactions depend critically on the reaction environment. Solvation, the local organization of the solvent molecules around a solute, has a central role in the description of condensed phase chemical properties. The influence of the solvent on the solute structure and reaction dynamics has been extensively studied[1–3], and the general success of continuum model descriptions bring into question the importance of short ranged molecular effects[4–6]. A diverse range of theoretical and simulation methods can decompose solvation dynamics and energetics into distinct short and long-range interactions, but few experiments provide robust tests of these models[7,8]. When the driving forces for solvation transition from physical to chemical interactions, the importance of site-specific interactions and dynamics becomes more prominent, as does the need for experimental probes that robustly sample the dynamics of solute–solvent interactions with atomic resolution. Here we present a femtosecond resolution hard X-ray scattering study of the electronic excited state dynamics of the model photocatalyst $[Ir_2(dimen)_4]^{2+}$ (dimen = diisocyano-para-menthane) in acetonitrile solution[9–13]. Figure 1a shows the molecular structure, and Fig. 1b shows a schematic representation of the experimental set-up.

The $[Ir_2(dimen)_4]^{2+}$ system belongs to a class of d8-d8 dimeric complexes, which include similarly bridged versions of Rh(I), Pt(II) and Os(0) dimers[9,14]. For these complexes, the lowest energy transition promotes an electron from an antibonding $\sigma^*_{d_{z^2}}$ to a bonding $\sigma_{p_z}$ orbital located in between the two metal atoms, increasing the formal bond order between the metal atoms from 0 to 1. Excitation thus leaves both the $\sigma^*_{d_{z^2}}$ and $\sigma_{p_z}$ orbitals partially occupied, making the excited state molecule both a better oxidant and reductant than the ground state molecule. Since both orbitals extend outside the molecule along the metal–metal axis, the metal atoms become open coordination sites for photoactivated reduction, oxidation and atom transfer reactions[9,10,15,16]. These photoreactions either directly involve the solvent, or happen in competition with active site coordination of the solvent. In general, the nature and dynamics of such active site solute–solvent interactions influence catalytic activity, but have been difficult to disentangle, because prior investigations typically relied on indirect methods such as optical spectroscopy with the atomistic interpretation being guided by molecular dynamics simulations[17,18].

Ultrafast X-ray diffuse scattering (XDS)[19–24] directly probes the time-dependent changes in the distribution of distances between all unique pairs of atoms[21,22,25–27], a property directly available from molecular dynamics simulations[13,22]. This makes the comparison between experiment and simulation straight forward by avoiding the often complex conversion of simulation results to spectroscopic observables. Additionally, ultrafast XDS accentuates significantly different dynamics than optical spectroscopies; XDS preferentially samples the dynamics associated with the most electron-rich atoms and directly probes bond distances and angles, while optical spectroscopy preferentially samples Franck-Condon active motions. Franck-Condon analysis suffers from the fact that these motions need not be chemically relevant and accurate electronic excited state potential surfaces are needed to extract structural information.

The properties of ultrafast diffuse X-ray scattering make the method an optimal approach to study the dynamics occurring locally around the photocatalytically active Ir atoms where the largest changes in intramolecular electronic and nuclear structure of $[Ir_2(dimen)_4]^{2+}$ occur[10]. Since the change in the XDS signal depends on the change in the interatomic distances between pairs of atoms and their electron density, the time resolved signal is very sensitive to changes in the structure surrounding the electron rich Ir atoms.

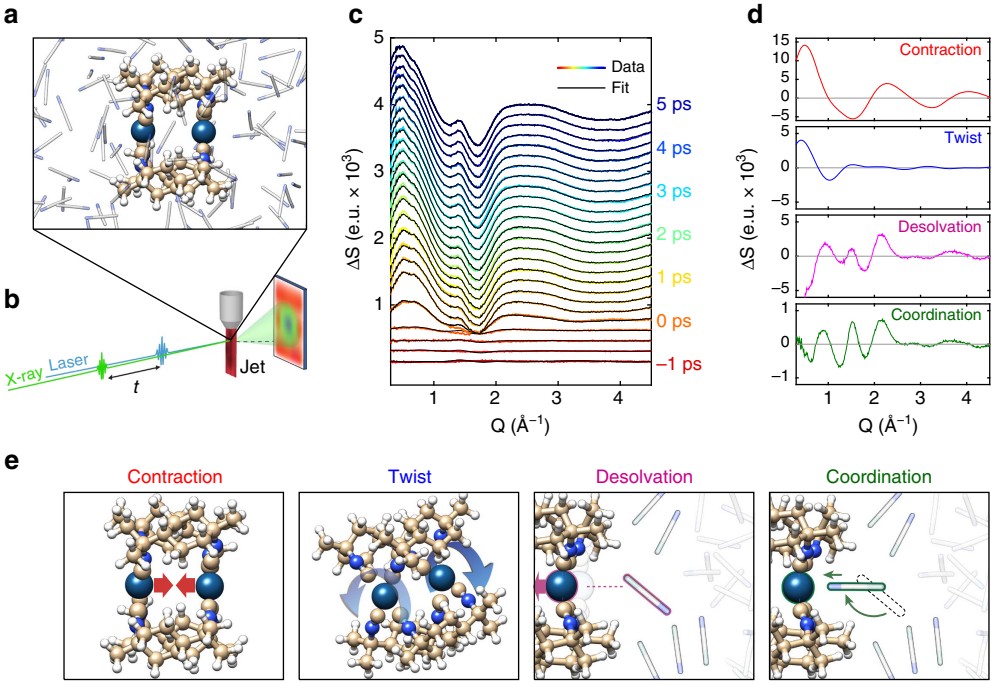

**Figure 1 | Scheme and results of the XDS experiments on $[Ir_2(dimen)_4]^{2+}$.** (**a**) Shows a snapshot of $[Ir_2(dimen)_4]^{2+}$ in acetonitrile solution from BOMD simulations. (**b**) Shows the experimental set-up. (**c**) Shows the recorded difference scattering data and fit, each consecutive curve has been offset by 150 e.u. for visibility. (**d**) Shows examples of the four components used to fit the data; The contraction signal is simulated for a 4.2 to 2.9 Å contraction of the Ir–Ir distance with no change in the ligand twist. The ligand twist component is simulated for a 0 to 15° degree increase in the N-Ir-Ir-N ligand dihedral twist at an Ir–Ir distance of 2.9 Å. The two solute components are extracted directly from the analysis. (**e**) Shows a sketch of four dynamics giving rise to the signals presented in **d**.

The ground state structure of $[Ir_2(dimen)_4]^{2+}$ excited with 480 nm light has a 4.3 Å Ir–Ir bond length and a $<5°$ N-Ir-Ir-N dihedral angle[11,28]. As photoexcitation promotes an electron from an antibonding to a bonding Ir–Ir orbital[9], the excitation is expected to be accompanied by a significant contraction along the newly formed Ir–Ir bond. A previous time resolved XDS study determined that the Ir–Ir distance contracts to 2.9 Å and the ligand dihedral twist increases by 15° in the metastable electronic excited state. However, the study did not resolve any intermolecular dynamics at the coordination site, and lacked the time resolution to resolve any intramolecular structural dynamics[11].

Herein, we apply XDS at an X-ray free-electron laser source to follow the dynamics of photo-excited $[Ir_2(dimen)_4]^{2+}$ in solution on the $\sim 100$ fs time scale of atomic movement within the solution. We thus monitor the signal from structural changes in the molecular system and its solvent shell as they happen. The experimental analysis of the XDS data is supported by Born-Oppenheimer Molecular Dynamics (BOMD) simulations. Both experiment and simulation provide direct access to the solute–solute and solute–solvent pair distribution functions, enabling robust characterization of both molecular structural dynamics and solvation dynamics around the catalytically active iridium sites.

## Results

**Experimental approach.** The present time-resolved diffuse X-ray scattering measurements were performed at the Linac Coherent Light Source X-ray Pump Probe instrument[29]. A 6 mM solution of $[Ir_2(dimen)_4]^{2+}$ dissolved in acetonitrile was pumped with 480 nm optical laser pulses of 70 fs duration, and the diffuse X-ray scattering generated by the $\sim 30$ fs X-ray laser probe pulses was recorded on an area detector[30] located behind the sample as shown in Fig. 1b. The optical laser induced difference signal as a function of time delay between pump and probe appears in Fig. 1c. The methods section contains a detailed description of the sample conditions, experimental set-up and the full data acquisition and reduction scheme[31].

The time dependent difference scattering signal $\Delta S(Q,t)$ is presented as coloured curves in Fig. 1c, where the momentum transfer $Q = \frac{4\pi \sin(\theta)}{\lambda}$ is determined by the scattering angle ($2\theta$) and X-ray wavelength ($\lambda$). The difference scattering signal arises from all structural dynamics induced by the laser pump pulse. In the standard analysis formalism[22,23,25], $\Delta S(Q,t)$ is described as the sum of difference scattering components arising from changes in the solute structure[32], changes in the solvation cage structure[33], and changes in the bulk solvent structure[34].

$$\Delta S(Q, t) = \Delta S_{\text{solute}}(Q, t) + \Delta S_{\text{solvation cage}}(Q, t) + \Delta S_{\text{bulk solvent}}(Q, t) \tag{1}$$

Each of these components can be further separated into specific molecular structural distortions and solvation processes. The difference scattering signal is analysed by simulating the contribution from each of these components and comparing the sum of these contributions to the data. In analysing $\Delta S(Q,t)$ (black curves Fig. 1c) we obtained a good description of the data using five components describing excited state dynamics of the recorded data: two solute components arising from changes in the Ir–Ir bond length ($d_{\text{Ir–Ir}}$), and N-Ir-Ir-N dihedral angle ($D_{\text{N-Ir-Ir-N}}$), two solvation cage components (an initial desolvation of the Ir atoms, followed by slower excited state coordination), and one solvent component arising from an increase in bulk solvent temperature ($\Delta T$) caused by energy dissipation from the

photo-excited solute molecules to the solvent.

$$\Delta S(Q, t) = \Delta S_{d_{\text{Ir-Ir}}}(Q, t) + \Delta S_{D_{\text{N-Ir-Ir-N}}}(Q, t) + \Delta S_{\text{desolvation}}(Q, t) + \Delta S_{\text{coordination}}(Q, t) + \Delta S_{\Delta T}(Q, t) \tag{2}$$

With the contribution of the temperature increase being $<5\%$ (see Supplementary Methods and Supplementary Figs 1–9), the difference scattering signal is dominated by the two solute and two solvation cage components, illustrated in Fig. 1d.

The difference scattering signal is dominated by a strong positive feature at low $Q$ indicative of decreasing interatomic distances[11,32]. This positive feature located at $Q \sim 0.6$ Å$^{-1}$ increases significantly during the first few hundred femtoseconds, followed by a sharpening and a shift to lower $Q$ on the picosecond timescale. The difference signal arising from the expected structural changes of the solute (Ir–Ir contraction and increasing dihedral twist) are presented in Fig. 1d, and both give rise to strong positive features at low $Q$. The Ir–Ir contraction gives rise to a broad positive feature at $Q = 0.55$ Å$^{-1}$, while the ligand twist deformation results in a sharper positive feature at $Q = 0.4$ Å$^{-1}$. The dynamics of the signal is thus consistent with the excited state initially undergoing a fast Ir–Ir contraction, followed by a slower N-Ir-Ir-N dihedral angle twist. The solvation cage dynamics and interconversion between ground state structures also contribute to the low $Q$ signal. Therefore, quantitative analysis of the difference scattering signal required an iterative analysis procedure, which is outlined in the following and described in detail in Supplementary Methods and shown in Supplementary Figs 10–13.

**Molecular structural dynamics.** Within the quantitative analysis of the XDS data, the scaled difference scattering is simulated for each time step and evaluated against the measured data. The signal arising from intramolecular structural dynamics is calculated directly from a large set of DFT-optimized molecular geometries where the Ir–Ir bond length and N-Ir-Ir-N dihedral angle have been systematically varied. Initial predictions of the solvation signal were extracted from quantum mechanical/molecular mechanical simulations, and the signal from the increase in bulk solvent temperature was taken from reference measurements[34]. Since a host of concurrent excited state structural dynamics are contributing to the difference scattering signal on early time scales, the analysis was optimized through a series of sequential steps, which ensured that overfitting of any one parameter was avoided.

In the following, we describe the results of the quantitative analysis of the difference scattering signal. Turning first to the intramolecular dynamics of $[Ir_2(dimen)_4]^{2+}$, Fig. 2a shows the measured time evolution of the average Ir–Ir contraction (red) and the average dihedral twist of the ligand system (blue), normalized to the deformations at 3.5 ps. Consistent with the qualitative conclusions above, we observe a significant Ir–Ir contraction in $<300$ fs, followed by an additional contraction on the 2 ps time scale. This contraction is observed to be modulated by a weak oscillatory feature. Simultaneously, we find the N-Ir-Ir-N dihedral angle twisting follows roughly one picosecond after the initial Ir–Ir bond contraction. The delayed dihedral twisting also occurs on a 2 ps time scale. For times exceeding 3.5 ps the fitting procedure returns an excited state structure ($d_{\text{Ir–Ir}} = 2.92 \pm 0.05$ Å and $\Delta\phi_{\text{N-Ir-Ir-N}} = 15 \pm 3°$) matching previous synchrotron measurements obtained with 50 ps resolution[11].

Recent quantum mechanical/molecular mechanical BOMD simulations of $[Ir_2(dimen)_4]^{2+}$ in acetonitrile[13] enable the robust interpretation of these experimental findings. Figure 2b shows the BOMD results, averaged over 40 trajectories, for the two

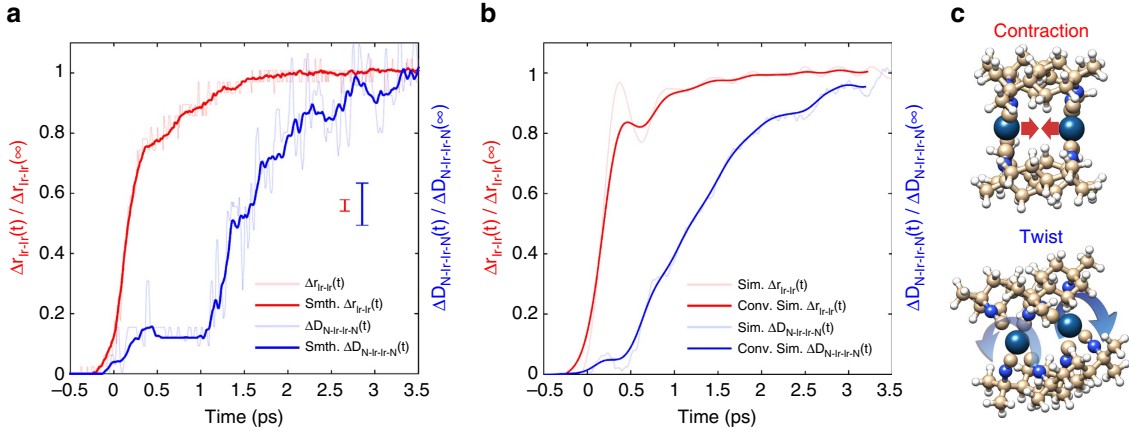

**Figure 2 | Molecular structural dynamics of photoexcited [Ir$_2$(dimen)$_4$]$^{2+}$.** (**a**) The Ir–Ir contraction (red) and the N-Ir-Ir-N dihedral twist (blue) determined from the XDS analysis. The full curves show the 15-point (~120 fs) smoothed result, with the parameter uncertainty being estimated by the point-to-point variation. Vertical lines show the average standard error at each time delay. (**b**) Ir–Ir contraction (red) and N-Ir-Ir-N dihedral twist (blue) determined from BOMD simulations (shaded lines) and the results from BOMD after convoluting the signal with the IRF of the experiment (full lines). (**c**) Schematic depiction of the contraction and twist distortions.

deformation modes, convolved with the time resolution of the experiment ($\sigma_{IRF} = 130$ fs, see Methods section). There is good agreement between experimental and simulated results (Fig. 2a,b), although the experimentally determined amplitude of the first oscillation of the Ir–Ir contraction is seen to be weaker than predicted by the BOMD simulations, while the oscillations appearing in the ligand twist deformation dynamics are more pronounced (see Supplementary Fig. 2 for a comparison of the experimental and simulated vibrational period). Taken together, simulation and measurement significantly expand our understanding of the intramolecular structural dynamics of [Ir$_2$(dimen)$_4$]$^{2+}$ compared with prior ultrafast optical spectroscopy measurements[12], which were insensitive to the delayed and diffusive dihedral angle deformation mode and did not provide quantitative results regarding bond lengths or angles.

**Solvation dynamics.** By first identifying the solute and bulk solvent contributions to the difference scattering signal, the residual difference signals provide a direct measure of the structural dynamics within the solvation shell. The 500 fs and 3 ps difference signals associated with the changes in the solvation shell structure are presented in Fig. 3a. These experimentally derived solvation shell signals can be directly compared with the results of the BOMD simulations. Figure 3c shows the difference scattering signal calculated[35] from the BOMD simulations at the same 500 fs and 3 ps time delays. The similarity between measured and calculated solvation shell signals provides strong support for guiding the interpretation of the measurements by the structural findings of the simulations. For both measured and simulated cage signals, the negative feature at $Q < 0.5$ Å$^{-1}$, which dominates the signal at 500 fs (Fig. 3a,c, red lines), has decreased significantly in intensity by the 3 ps time delay (Fig. 3a,c, blue lines). The negative feature at low $Q$ arises from the increase in average distance between the solvating acetonitrile molecules and the Ir atoms due to the inward motion of the Ir atoms upon photoexcitation. The BOMD simulations show that in the ground state, the Ir atoms are preferentially solvated by methyl groups held in place through electrostatic interactions between electronegative ligand nitrogen atoms and the acetonitrile methyl groups (see Supplementary Methods and Supplementary Figs 14–16). We conclude this sub-ps process corresponds to methyl group desolvation of the Ir atoms (Fig. 3d, red line) as a consequence of photo-initiated Ir–Ir bond contraction. On the

few-picosecond time scale, the solvent cage adapts to the structure and electronic configuration of the excited state [Ir$_2$(dimen)$_4$]$^{2+}$. The hole in the $\sigma^*_{d_{z^2}}$ orbital created by photoexcitation makes Ir a stronger Lewis acid and gives rise to a specific coordination of the Ir atoms by the N lone pair electrons of acetonitrile (Fig. 3d, blue line). In the following, we model the experimental solvation dynamics with the 500 fs and 3 ps solvation cage difference signals as signatures of two distinct solvation processes. This model provides an accurate representation of the experimental data at all time delays (see Fig. 1c). The initial desolvation corresponds to the preferential loss of acetonitrile methyl group solvation of the Ir atoms. This loss does not recover, but rather the acetonitrile molecules rotate and translate to preferentially coordinate the Ir atoms with the cyano group. The time dependent evolution of these solvation processes determined from the experiment appear in Fig. 3b. This descriptive interpretation of the experiment relies on the qualitative agreement between the experimental and BOMD solvent cage signals shown in Fig. 3a,c.

A schematic representation of the two distinct solvation processes appears in Fig. 3e,f. The delayed onset of the nitrogen coordination observed both experimentally and in the BOMD simulations reflects the large amplitude rotations and translations acetonitrile molecules must go through to coordinate the electronic excited state relative to the ground state. The active site coordination processes of the excited state [Ir$_2$(dimen)$_4$]$^{2+}$ (blue trace in Fig. 3b) is well-described by a 1.3 ps delay followed by a 2.0 ps single-exponential rise (see Supplementary Fig. 1). While both coordination and dihedral angle twisting are delayed with respect to the excitation event, the non-equilibrium coordination process is significantly slower. The primary difference between the measured and simulated cage response (Fig. 3a,c) pertains to the dip and peak feature at 1.2 and 1.6 Å$^{-1}$. As described in 'Supplementary Methods and shown in Supplementary Figs 5–9, the strength of this $Q$-dependent modulation can be correlated to changes in the angle between the coordinating nitrile group and the Ir–Ir axis indicating that the BOMD simulations underestimate the axial alignment between coordinating acetonitrile and the Ir–Ir axis in the excited state. Asides from this deviation, all experimental predictions on the solvation dynamics provided by the BOMD calculations (both in terms of overall signal shape and temporal evolution) have been identified in the XDS data. Thus while XDS data alone does not provide the ability to differentiate between the cyano nitrogen

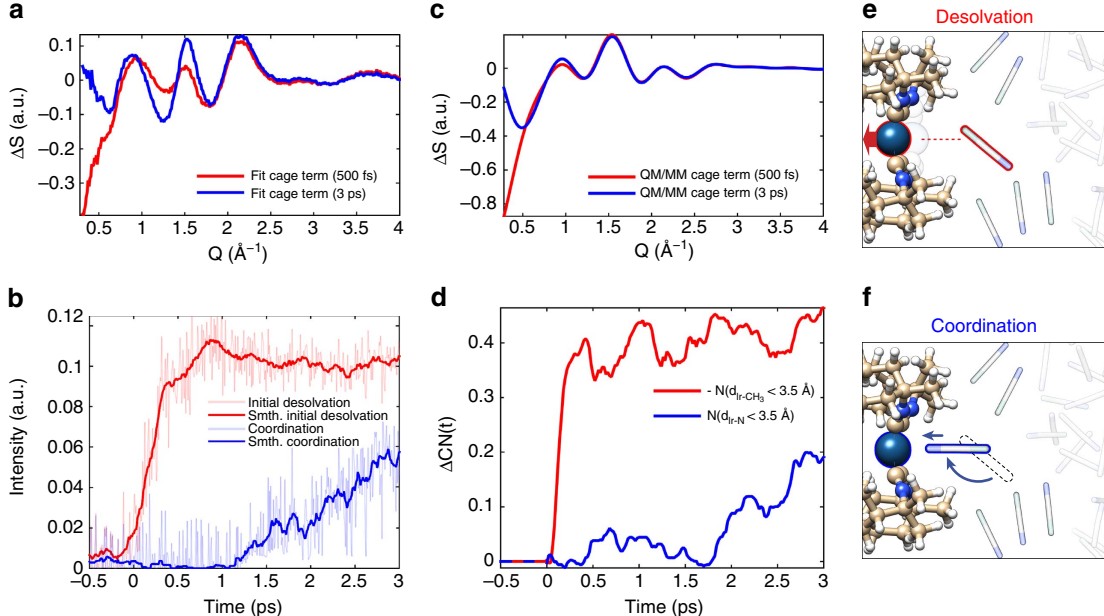

**Figure 3 | Solvation dynamics of photoexcited $[Ir_2(dimen)_4]^{2+}$ in acetonitrile.** (**a**) Solvation cage signal extracted from the difference scattering data recorded at time delays of 500 fs and 3 ps. (**b**) Evolution of the two experimentally determined solvation components. Full curves show the 15-point smoothed result, with the uncertainty being estimated by the point-to-point variation in the fit shown as transparent curves. (**c**) Simulated difference scattering from the solvent cage of the BOMD simulations for time delays of 500 fs and 3 ps. Simulated signal has been scaled by the 13% excitation fraction to facilitate comparison with cage term extracted from data. (**d**) Change in coordination number (CN) of the Ir atoms, by the methyl and nitrogen groups of the acetonitrile solvent after the excitation as predicted by the BOMD simulations (note that the methyl coordination change is negative). (**e,f**) Schematic depiction of the processes giving rise to the solvation signal measured in the difference scattering data; (**e**) An initial loss of solvation of the Ir atoms of $[Ir_2(dimen)_4]^{2+}$ by the methyl groups of the acetonitrile solvent and (**f**) a coordination of the Ir atoms of the excited state $[Ir_2(dimen)_4]^{2+}$ by the nitrogen groups of the acetonitrile solvent.

and methyl solvation, such mechanistic insights and atomic specificity can be provided by the inclusion of statistically robust BOMD simulations in the XDS analysis.

## Discussion

The observed solvation dynamics in acetonitrile differ significantly from those measured with more traditional ultrafast optical probing of solvation dynamics initiated by photoexcitation of organic dyes. The archetypical polar solvation dynamics in acetonitrile extracted from ultrafast optical spectroscopy observe dynamics dominated by the rotational response of acetonitrile with the slowest dynamics occurring on a sub-ps time scale[4,36,37]. This well-established picture does not apply to $[Ir_2(dimen)_4]^{2+}$ because the perturbation induced by photo-excitation of $[Ir_2(dimen)_4]^{2+}$ does not change the dipole moment of the complex but significantly changes the local solvation structure surrounding the Ir atoms. Transitioning from the solvation structure of the electronic ground state to that of the excited state requires large translational motions strongly coupled to rotations, leading to much slower dynamics than those associated with the standard picture of solvation in acetonitrile. By combing ultrafast XDS with BOMD simulation, we have been able to characterize and assess these complex structure dynamics with atomic-site specificity. We believe that the combined use of statistically robust BOMD simulations and ultrafast XDS will provide a powerful approach to the further development of a truly molecular-scale view of solvation. Such molecular-scale atomistic information, can be supplemented by element specific characterization of excited state electronic configuration with X-ray emission and X-ray absorption measurements, which can be recorded simultaneously with XDS[24,33]. The combination of these complementary X-ray methods presents the opportunity to map photocatalysis processes with unprecedented detail.

## Methods

**Materials.** The 6 mM $[Ir_2(dimen)_4]^{2+}$ sample was synthesized by direct mixing of 1,8-diisocyano-paramenthane (dimen) and $Ir_2Cl_2(COD)_2$ (COD = 1,5-cyclooctadiene) in a 4:1 ratio in degassed acetonitrile in a glovebox. The sample integrity was monitored by continuously measuring the absorption spectrum during the X-ray experiment.

**Experimental X-ray set-up.** The time-resolved scattering measurements were performed at the X-ray pump-probe instrument at the Linac Coherent Light Source[29].

The sample, consisting of 6 mM $Ir_2(dimen)_4^{2+}$ in acetonitrile solution, was pumped though a closed-loop system with a 100 μm nozzle (Kyburtz) producing a fast-flowing, flat sample sheet in the sample/beam interaction region with full sample refresh between successive pump/probe events (120 Hz). A gear pump ensured continuous refresh of the probed sample volume from a 80 ml reservoir kept under He atmosphere. The nozzle and catcher were housed in a He-filled aluminum sample chamber with a kapton-covered window allowing the scattered X-ray photons to exit the chamber. The scattered X-rays were collected at scattering angles between 3 and 70 degrees corresponding to a Q-range of 0.2 to 4.5 Å$^{-1}$ utilizing the Cornell-SLAC Pixel Array Detector (CSPAD v1.2) (ref. 38). The sample was excited with 480 nm vertically polarized laser pulses, with ∼70 fs FWHM duration, generated by an OPA from the 800 nm fundamental of a Ti:sapphire laser. The optical laser spot size was 255 × 255 μm (FWHM) and the power was 25 μJ per pulse measured at the sample position.

The 9.5 keV ∼0.3% bandwidth X-ray pulses were used to probe the sample after pumping with the optical laser. The time delay, $\Delta t$, of the arrival of the optical laser relative to the X-ray probe was varied in order to monitor the change in the scattering signal as a function of the delay.

The shot-to-shot fluctuations in the relative timing between X-ray and optical pulses were monitored for each pump/probe event using the so-called Timing Tool for timing diagnostic. This timing diagnostic tool is based on optical detection of X-ray generated carriers in a $Si_3N_4$ thin film, and is described in detail elsewhere[39].

**Data treatment.** The shot-to-shot fluctuations in X-ray energy and intensity and the resulting detector response was characterized and subtracted[31] as well as corrected for spatial corrections such as polarization, solid angle coverage and absorption through the liquid sheet. Finally, difference scattering curves are produced as a function of wavevector transfer and time delay, see Supplementary Methods.

The instrument response function (IRF) of the experiment was estimated by convoluting the kinetics extracted for the change in Ir–Ir distance estimated through the BOMD calculations with a Gaussian response function and then minimizing the residual between the simulated and measured contraction as a function of this IRF width. This procedure resulted in an IRF with $\sigma = 130 \pm 20$ fs.

An IRF of 130 fs is in good agreement with the expected $\sim 1$ fs $\mu m^{-1}$ temporal broadening due to the combination of the velocity mismatch between the pump- and probe pulses as they traverse the 100 um liquid jet, the 70 fs FWHM of the laser pulse and the 30 fs FWHM of the X-ray pulse, resulting in an expected IRF of 125 fs.

**Data availability.** All relevant data are available from the authors upon request.

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

## Acknowledgements

T.B.v.D., K.S.K., A.O.D., M.C., K.H. and M.M.N. acknowledge support from DANSCATT. K.S.K., K.B.M. and M.M.N. acknowledge support from the Danish Council for Independent Research. A.O.D., K.B.M., M.M.N. and N.E.H. acknowledge support from the Lundbeck Foundation. K.S.K. acknowledges support from the Carlsberg Foundation. W.G. acknowledges support from the European X-ray free-electron laser and the Deutsche Forschungsgemeinschaft (D.F.G.) via SFB 925 (project A4). R.W.H., Z.S., W.Z. and K.J.G. acknowledge support from the AMOS programme within the Chemical Sciences, Geosciences and Biosciences Division of the Office of Basic Energy Sciences, Office of Science, US Department of Energy. Portions of this research were carried out at the Linac Coherent Light Source, a national user facility operated by Stanford University on behalf of the US Department of Energy, Office of Basic Energy Sciences. J.G.K., K.H.K. and H.I. acknowledge support from Institute for Basic Science (IBS-R004-G2).

## Author contributions

R.W.H., T.B.v.D., K.S.K., M.M.N., K.H., and K.J.G. designed the X-ray experiment. T.H. synthesized the samples. T.B.v.D., K.S.K., R.W.H., A.O.D., T.H., Mo.C., Ma.C., W.G., J.G.K., K.H., K.H.K., J.K., H.L., Z.S, W.Z., D.Z., M.M.N. and K.J.G. conducted the X-ray experiment. A.O.D., T.B.v.D., K.B.M. and N.E.H. conducted the computational chemistry. T.B.v.D. and K.S.K. analysed the data. K.S.K., T.B.v.D., A.O.D. and K.J.G. wrote the manuscript. All authors discussed the results and commented on the manuscript.

## Additional information

**Competing financial interests:** The authors declare no competing financial interests.

**Publisher's note**: 

