## [Peer Review File · Nature Communications]

REVIEWERS' COMMENTS:

Reviewer #1 (Remarks to the Author):

The revised version of the paper by van Driel et al. and the authors' reply letter exhaustively address my comments and questions on the previous version of the manuscript. I think that the paper is now suitable for publication in Nature Communication, subjected to the minor points noted below.

Indeed, although the light-driven dynamics in the $[\text{Ir}_2(\text{dimen})_4]^{2+}$ complex has been already investigated in detail with both X-ray and non-X-ray methods, the new insights on ultrafast solvation dynamics and the role of BOMD simulations in the interpretation of the XDS data represent a proof-of-concept advance of significance to the experimental and theoretical communities working with solution phase photochemistry at XFELs. The well-organized SI material could be also a very useful guide for new users approaching the technique.

I'm very much looking forward to the next big step with these methods, capitalizing on the accumulated background to move from model systems to actual photocatalysts in action.

Minor points:

- 1) In Fig. 1D the ordinate axis scale label perhaps should be "(e.u.) 10^3 " and not "(e.u.) $10^{\{-3\}}$ ". Please check. Also, in Fig. 1C a bar indicator for the ordinate axis scale would be useful.
- 2) In the sentence: "... the analysis was optimized through a series sequential steps, which ensured that overfitting of any one parameter was avoided" an "of" is likely missing before "sequential steps".
- 3) Fig. 1 was wrongly repeated towards the end of the manuscript, after the sentence: "a coordination of the Ir atoms of the excited state $[\text{Ir}_2(\text{dimen})_4]^{2+}$...".
- 4) I think that the conclusions of the paper could benefit from a brief comment on the use of other ultrafast X-ray based techniques already demonstrated at XFELs, such as X-ray Absorption and Emission spectroscopies. These element-selective spectroscopies in principle would be highly complementary to XDS, especially with respect to atomic-site specificity in the analysis of light-driven structural dynamics.
- 5) I appreciated that difference signals, fit and fit components at the different time delays probed have been added in SI as Figure S13. However, for additional clarity and consistency with the rest of the Figures, I suggest to express also here the XDS signal in electron units (e.u.) instead of (a.u.).

Reviewer #2 (Remarks to the Author):

This reviewer believes that the authors have addressed the concerns raised by the first round of reviewer and have answered questions and made corrections accordingly. Therefore, I have no further comments regarding its publication.

Reviewer #1 (Remarks to the Author):

The revised version of the paper by van Driel et al. and the authors' reply letter exhaustively address my comments and questions on the previous version of the manuscript. I think that the paper is now suitable for publication in Nature Communication, subjected to the minor points noted below.

Indeed, although the light-driven dynamics in the $[\text{Ir}_2(\text{dimen})_4]^{2+}$ complex has been already investigated in detail with both X-ray and non-X-ray methods, the new insights on ultrafast solvation dynamics and the role of BOMD simulations in the interpretation of the XDS data represent a proof-of-concept advance of significance to the experimental and theoretical communities working with solution phase photochemistry at XFELs. The well-organized SI material could be also a very useful guide for new users approaching the technique. I'm very much looking forward to the next big step with these methods, capitalizing on the accumulated background to move from model systems to actual photocatalysts in action.

- *We thank the reviewer for the positive comments.*

Minor points:

1) In Fig. 1D the ordinate axis scale label perhaps should be "(e.u.) 10^3 " and not "(e.u.) 10^{-3} ". Please check. Also, in Fig. 1C a bar indicator for the ordinate axis scale would be useful.

- *We have updated Figure 1c and 1d accordingly.*

2) In the sentence: "... the analysis was optimized through a series sequential steps, which ensured that overfitting of any one parameter was avoided" an "of" is likely missing before "sequential steps".

- *The reviewer is right. We have added an 'of'.*

3) Fig. 1 was wrongly repeated towards the end of the manuscript, after the sentence: "a coordination of the Ir atoms of the excited state $[\text{Ir}_2(\text{dimen})_4]^{2+}$...".

- *The excess Figure 1 has been removed...*

4) I think that the conclusions of the paper could benefit from a brief comment on the use of other ultrafast X-ray based techniques already demonstrated at XFELs, such as X-ray Absorption and Emission spectroscopies. These element-selective spectroscopies in principle would be highly complementary to XDS, especially with respect to atomic-site specificity in the analysis of light-driven structural dynamics.

- *We have added the following sentence to the discussion:*

"Such molecular-scale atomistic information, can be supplemented by element specific characterization of excited state electronic configuration with x-ray emission and x-ray absorption measurements, which can be recorded simultaneously with XDS^{Error! Reference source not found.,Error! Reference source not found.}. The combination of these complementary x-ray methods presents the opportunity to map photocatalysis processes with unprecedented detail."

5) I appreciated that difference signals, fit and fit components at the different time delays probed have been added in SI as Figure S13. However, for additional clarity and consistency with the rest of the Figures, I suggest to express also here the XDS signal in electron units (e.u.) instead of (a.u.).

- *We have updated the figures to (e.u.).*

Reviewer #2 (Remarks to the Author):

This reviewer believes that the authors have addressed the concerns raised by the first round of reviewer and have answered questions and made corrections accordingly. Therefore, I have no further comments regarding its publication.

- *Great.*